



# Coupled online learning as a way to tackle instabilities and biases in neural network parameterizations: general algorithms and Lorenz 96 case study (v1.0)

Stephan Rasp[1]

[1]Technical University of Munich, Germany

**Correspondence:** Stephan Rasp (stephan.rasp@tum.de)

**Abstract.** Over the last couple of years, machine learning parameterizations have emerged as a potential way to improve the representation of sub-grid processes in Earth System Models (ESMs). So far, all studies were based on the same three-step approach: first a training dataset was created from a high-resolution simulation, then a machine learning algorithms was fitted to this dataset, before the trained algorithms was implemented in the ESM. The resulting online simulations were frequently plagued by instabilities and biases. Here, coupled online learning is proposed as a way to combat these issues. Coupled learning can be seen as a second training stage in which the pretrained machine learning parameterization, specifically a neural network, is run in parallel with a high-resolution simulation. The high-resolution simulation is kept in sync with the neural network-driven ESM through constant nudging. This enables the neural network to learn from the tendencies that the high-resolution simulation would produce if it experienced the states the neural network creates. The concept is illustrated using the Lorenz 96 model, where coupled learning is able to recover the "true" parameterizations. Further, detailed algorithms for the implementation of coupled learning in 3D cloud-resolving models and the super parameterization framework are presented. Finally, outstanding challenges and issues not resolved by this approach are discussed.

## 1 Introduction

The representation of subgrid processes, especially clouds, is the main cause of uncertainty in climate projections and a large error source in weather predictions (Schneider et al., 2017b). Models that explicitly resolve the most difficult processes are now available but are too expensive for operational forecasting. Machine learning (ML) has emerged as a potential shortcut which would allow using short-term high-resolution simulations in order to improve climate and weather models. However, two issues have plagued all approaches so far: First, simulations with neural networks turned out to be unstable at times. Second, even if stable, the resulting simulations had biases compared to the reference model. In pre-ML climate model development, biases were reduced by manual tuning of a handful of well-known parameters (Hourdin et al., 2017). With thousands of non-physical parameters in a neural network, this is no longer possible. In this paper, I propose coupled online learning as a potential





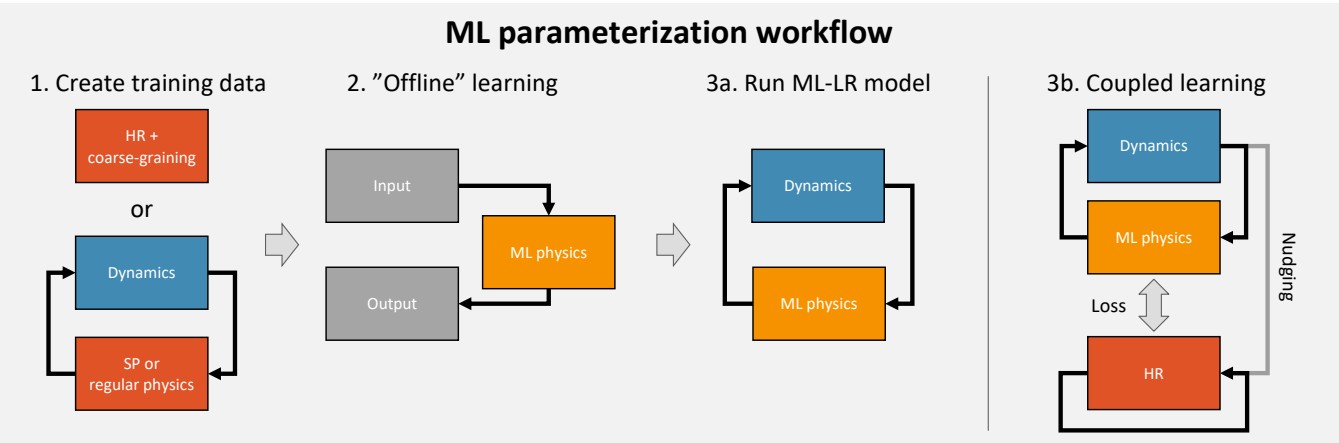

**Figure 1.** Schematic overview of ML parameterization workflow with and without coupled online learning.

mechanism to tackle these two issues and illustrate the principle using the two-level Lorenz 96 (L96) model, a common (but probably too simple) model of multi-scale atmospheric flow (Lorenz, 1995)[1].

## 25  2  Review of online machine learning parameterizations

Over the last couple of years, several attempts have been made at building ML subgrid parameterizations, all of which followed a similar approach (Fig. 1). The first step is to create a training dataset from a reference simulation. In step two, this dataset is then used to train a ML algorithm. After training, the predictions of the algorithm can then be compared offline against a validation dataset. Promising offline results have been obtained for a number of subgrid processes (Krasnopolsky et al.,

2013; Bolton and Zanna, 2019). Step three is to implement the ML algorithm in the climate model code where it replaces the traditional subgrid schemes and is coupled the the dynamical core and non-ML parameterizations. These hybrid models are then integrated forward in what I will call online mode. So far, three studies have implemented all three steps in the context of an atmospheric model (Brenowitz and Bretherton, 2018; O'Gorman and Dwyer, 2018; Rasp et al., 2018). Note that all of these studies used a simplified aquaplanet world and the ML parameterizations only included the most important variables in their

input/output vectors. Cloud water and ice, for example, were omitted for the sake of simplicity.

### 2.1  Rasp et al. (2018) – Super-parameterization with a neural network

The three attempts differ in training data and ML algorithms used. In Rasp et al. (2018)(RPG18), we used a super-parameterized climate model as our training model (Khairoutdinov and Randall, 2001). In super-parameterization (SP), a 2D cloud-resolving model (CRM; $\Delta x = 4\,\mathrm{km}$) is embedded in each global circulation model (GCM; $\Delta x \approx 200\,\mathrm{km}$, $\Delta t = 30\,\mathrm{min}$) grid column. The

---

[1]Confusingly, even though the paper appears to have been published in 1995, most people refer to the model as the Lorenz 96 model.





CRM handles convection, turbulence and microphysics, while radiation[2], surface processes and the dynamics are computed on the GCM grid as usual. Compared to a global 3D CRM, SP is obviously less realistic but has several conceptual and technical advantages. First, sub-grid and grid scale processes are clearly separated, which makes it easy to define the parameterization task for a ML algorithm. Second, because the CRM lives in isolation, it exactly conserves certain quantities (e.g. energy and mass). A third, very practical advantage is that SP simulations are significantly cheaper than global 3D CRMs. In our study we trained a deep neural network to emulate the CRM tendencies. The offline validation scores were very encouraging (Gentine et al., 2018) even though the deterministic ML parameterization was unable to reproduce the variability in the boundary layer. In our subsequent online tests, we managed to engineer a stable model that produced results close to the original SP-GCM. However, small changes, either to the training dataset or in the input and output vectors, quickly led to unpredictable blow-ups. In these cases the network would output increasingly unrealistic tendencies at individual grid columns. Further, some biases to the reference model were evident (Fig. 1 in RPG18).

### 2.2 Brenowitz and Bretherton (2018) – Global 3D CRM with a neural network

Brenowitz and Bretherton (2018)(BB18) (extended in Brenowitz and Bretherton (2019)) used a 3D CRM ($\Delta x = 4\,\text{km}$, $\Delta t = 10\,\text{s}$) to create their reference simulation. This requires an additional spatial and temporal coarse-graining step to generate the training data for a ML parameterization for a coarser resolution model (in their case $\Delta x = 160\,\text{km}$, $\Delta t = 3\,\text{h}$). The challenge is to find the apparent subgrid tendencies. BB18 computed the subgrid tendency $\left(\partial\bar{\phi}/\partial t\right)_{sg}$ of an arbitrary variable $\phi$ (e.g. temperature or humidity) as the residual of the total coarse-grained tendency and the coarse-grained advection term:

$$\underbrace{\frac{\partial\bar{\phi}}{\partial t}}_{\text{Total coarse-grained tendency}} + \underbrace{\overline{\mathbf{v}}\cdot\overline{\nabla}\bar{\phi}}_{\text{Coarse-grained advection}} = \left(\frac{\partial\bar{\phi}}{\partial t}\right)_{sg} \tag{1}$$

This coarse-graining procedure assumes that the coarse-grained advection term closely resembles the dynamics of the coarse-grid GCM. This assumption might not be true in many cases. Further, the residual "sub-grid" terms do not obey any conservation constraints.

BB18 then fitted a neural network to the coarse-grained data, which produces good results in offline mode. In online mode, however they also experienced instabilities. Brenowitz and Bretherton (2019) identified unphysical correlations learned by the network as the cause for the instabilities and used two fixes to produce stable longer-term simulations. The first fix is to cut off upper levels from the input vector. The second fix involves an multi-time-step loss-function that integrates the network predictions forward in a single-column model setup. This essentially penalizes unstable feedback loops. Despite these improvements, the simulation drifts, potentially as a result of the coarse-graining issues mentioned in the previous paragraph.

---

[2]In some SPCAM versions radiation is computed on the CRM grid.





### 2.3 O'Gorman and Dwyer (2018) – Traditional parameterization with a random forest

The third online parameterization by O'Gorman and Dwyer (2018) uses a traditional parameterization as reference. For cloud parameterizations, this is mainly a proof-of-concept. For other, computationally expensive parameterizations, such as line-by-

line radiation parameterizations, pure ML emulation is a promising target. As with our super-parameterization, this way the parameterization task is clearly defined. The main difference of O'Gorman and Dwyer (2018) to RPG18 and BB18 is the ML method: a random forest (Breiman, 2001). Rather than learning a regression, as neural networks do, random forests essentially learn a multi-dimensional lookup table. Advantages of this approach are: 1) The predictions of a random forest are limited by what it has seen in the training dataset. This means it cannot produce "unphysical" tendencies which could lead to model blow-

ups. 2) Since the training data obeys physical constraints, so will the random forest predictions by default.[3] Random forests are also competitive with neural networks for many types of ML problems. In this paper, however, I will not further discuss random forests, since they are do not lend themselves to incremental online learning in their most common implementations. Note, however, that there are online learning algorithms for random forests (Saffari et al., 2009).

### 3 Coupled online learning – the general concept

Coupled online learning is essentially a second training step after the first offline training on a reference dataset. The basic idea of coupled learning is to run the ML-LR model in parallel with the HR model and train the network every or every few time steps (3b. in Fig. 1)[4]. The HR model is continuously nudged towards the LR model state keeping the two simulations close to each other. How close the two runs are depends on the nudging time scale $\tau_{\text{nudging}}$. A small nudging time scale forces the models closer together but the HR model might respond unrealistically or eventually blow up if the nudging is too strong.

Assuming that the HR and LR model state are close together, this method allows the ML parameterization to see what the HR model would do if it lived in the ML-LR model world. This should help in reducing biases and preventing instabilities. Take as an example a neural network parameterization that develops an unstable feedback loop and starts producing highly unrealistic tendencies. With offline learning only, the model will eventually blow up. In coupled learning, such unrealistic predictions would result in large losses. In the next gradient descent step the network will learn not to produce such tendencies any more.

The hope is that during this coupled learning phase the errors of the network will become smaller and smaller, so that eventually the ML-LR model can be run without supervision. Ideally, one could intermittently turn on the "supervising" HR model for cases where the ML parameterization starts to produce undesired tendencies. However, one has to consider that HR models require a spin up phase, which prohibits immediate deployment from a cold start. This problem might be less pronounced in

---

[3]At least to a good degree of approximation. Predictions of decision trees and therefore also random forests are averages over several training targets. Each target will perfectly obey constraints. Since the conservation constraints are likely non-linear, an average does not necessarily keep this property but probably comes close.

[4]A note on the terminology: I will use the terms HR (high-resolution) and LR (low-resolution) here when speaking about the general algorithm. When talking specifically about atmospheric science applications, I will use the more common terms CRM and GCM.

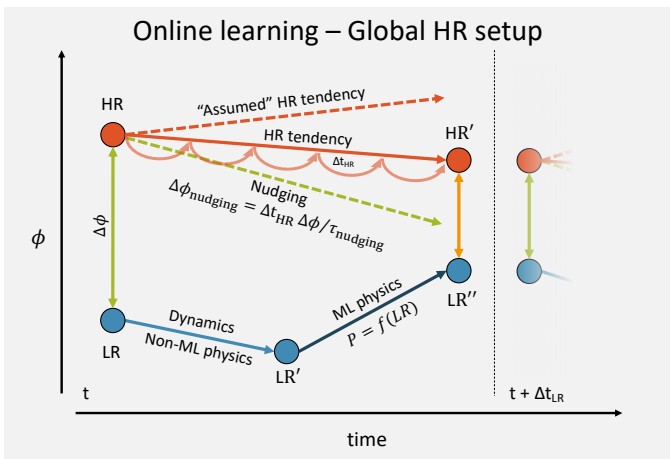

**Figure 2.** Evolution of a tracer $\phi$ during one LR model time step. This schematic applies the the L96 and 3D HR model case.

the case of an embedded HR model (as in SP) but nevertheless motivates the approach in this paper of continuously running
the two models in parallel.

The instability issues in previous studies can also be seen as a consequence of overfitting to the reference simulation used for
training. Once the ML parameterization is coupled to the LR model it will create its own climate which likely lies somewhat
outside the training manifold. This can easily lead to problems because neural networks struggle to extrapolate beyond what
they have seen during training. Coupled learning combats this problem by extending the training with HR targets for each state
that the ML-LR model produces.

The algorithmic details of coupled learning differ depending on the exact model setup. The main contribution of this paper
will be to describe coupled learning algorithms for the simple L96 model as well as global 3D HR models and SP models. To
understand how coupled learning actually works it is helpful to draw diagrams for the evolution of a tracer $\phi$ (e.g. temperature)
at one grid point during one LR model time step. I will start with the case of the 3D HR setup, which also applies to the
L96 model (Fig. 2). At the beginning of the time step $\phi$ will generally have different values in the LR and HR model (the
HR values are coarse-grained to the LR model grid). The difference $\Delta\phi = $ LR - HR is used to compute the nudging tendency
$\Delta\phi/\tau_{\mathrm{nudging}}$ during the HR model integration. The total increment from nudging then is $\Delta\phi_{\mathrm{nudging}} = \Delta\phi/\tau_{\mathrm{nudging}} * \Delta t_{\mathrm{LR}}$. In
addition to the nudging increment, the HR model also evolves on its own. Assuming that during this short time interval the
nudging and the HR-internal evolution are independent, the state of the HR model at the end of the LR model time step (HR$'$)
is a linear superposition[5]. The "assumed" HR-internal increment can be computed as $\Delta\phi_{\mathrm{HR\text{-}internal}} = $ HR$'$ $-$ HR $- \Delta\phi_{\mathrm{nudging}}$. In
the meantime, the LR model will first execute its dynamical core and any other parameterizations that are not represented by
a ML algorithm[6]. The resulting state is LR$'$. Then the ML parameterization will be called and the resulting tendencies will be

---

[5]I will call this the *linear superposition assumption* in the rest of the paper.

[6]Typically, in a LR model time step the physics is run before the dynamics. But where the time step starts and ends is arbitrary, so the two can be switched
without problems.





added to give LR$''$ = LR$'$ + $\mathcal{P}$(LR). One open question is whether the input to the ML parameterization should be LR or LR$'$. In this study LR is used but the differences are small. If the ML-LR model was a perfect emulation of the HR model, the total

LR increment LR$''$ − LR should be equal to the HR increment $\Delta\phi_{\text{HR-internal}}$. Therefore, the target for the parameterization is $y = \Delta\phi_{\text{HR-internal}} - (\text{LR}' - \text{LR})$. and the mean squared error loss is $\mathcal{L} = \frac{1}{N}\sum_i(y - \mathcal{P}(\text{LR}))^2$. The ML parameterization is then optimized every few time steps.

## 4   Parameterization experiments using the Lorenz 96 model

### 4.1   The L96 model

The L96 model is an idealized model of atmospheric circulation that, in its two-level variant, has been extensively used for parameterization research (Wilks, 2005; Crommelin and Vanden-Eijnden, 2008). Here, I use the model as described in Schneider et al. (2017a). Briefly, the model consists of a slow variable $X_k$ ($k = 1, \ldots, K$) and a coupled fast variable $Y_{j,k}$ ($j = 1, \ldots, J$):

$$\frac{dX_k}{dt} = \underbrace{-X_{k-1}(X_{k-2} - X_{k+1})}_{\text{Advection}} \underbrace{-X_k}_{\text{Diffusion}} \underbrace{+F}_{\text{Forcing}} \underbrace{-hc\bar{Y}_k}_{\text{Coupling}} \tag{2}$$

$$\frac{1}{c}\frac{dY_{j,k}}{dt} = \underbrace{-bY_{j+1,k}(Y_{j+2,k} - Y_{j+1,k})}_{\text{Advection}} \underbrace{-Y_{j,k}}_{\text{Diffusion}} \underbrace{+\frac{h}{J}X_k}_{\text{Coupling}} \tag{3}$$

Both, $X$ and $Y$ are periodic. $K = 36$, $J = 10$, $h = 1$ and $F = c = b = 10$. These parameters indicate that the fast variable evolves 10 times faster than the slow variable and has one tenth of the amplitude. A Runge-Kutta 4th order scheme with a time step of 0.001 is used to integrate these equations. The one-level model consists only of equation 2 without the coupling term on the right hand side[7].

For parameterization research, $X$ represents the large-scale, resolved variables, whereas $Y$ represents the small-scale, unre-

solved variables. The job of a parameterization $\mathcal{P}$ is to approximate the coupling term in the $X$ equation:

$$-hc\bar{Y}_k := B_k \approx \mathcal{P}(X_k) \tag{4}$$

Here, I only consider deterministic parameterizations that are local in space and time. The parameterization task is shown in Fig. 3.

### 4.2   Machine learning parameterizations

Two parameterizations will be considered: a linear regression and a neural network. The linear regression case is easily interpretable and helps to illustrate the learning procedure, while the neural network is a more realistic case.

---

[7]For animations of the L96 system, see https://raspstephan.github.io/blog/lorenz-96-is-too-easy/



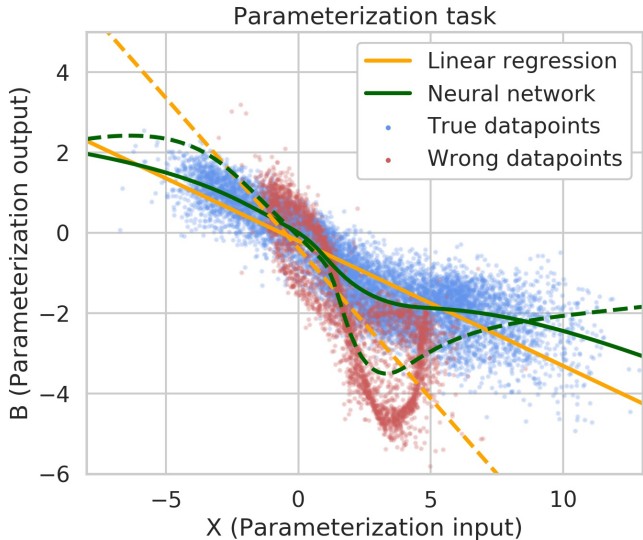

**Figure 3.** Blue dots are data points from a reference simulation with the real L96 parameters. The solid orange and green lines are the linear regression and neural network parameterization fitted to this data. The red dots are data points from the L96 simulations with "wrong" parameter values used for pretraining. The dashed lines are the parameterization fits for these "wrong" values, which serve as a starting point for the coupled learning experiments.

The linear regression parameterization looks as follows:

$$B_k = aX_k + b \tag{5}$$

When fitted to the points shown in Fig. 3, $a = -0.31$ and $b = -0.20$.

Neural networks consist of one or multiple layers of linearly connected nodes, modified by non-linear activation functions.[8] Here, I use a neural network with 2 hidden layers of 32 nodes in-between the input and output layer, which both have size 1. The total number of parameters is 1,153. The hidden layers are passed through an exponential linear unit (ELU) activation function. A neural network fit to real data is also shown in Fig. 3.

### 4.3 Coupled online learning[9]

To mimic the situation in a real climate model where the parameterization would first be pretrained *offline* on a traditional parameterization, super-parameterization or coarse-grained dataset, a training dataset using the full L96 equations but with different parameters was created: $F = 7$, $h = 2$, $c = b = 5$. The resulting, "wrong" data points along with the linear regression and neural network parameterizations are also shown in Fig. 3.

---

[8]For a great introduction to neural networks, see Nielsen (2015)

[9]All experiments were done in a Jupyter notebook that can be launched via Binder from the GitHub repository: https://github.com/raspstephan/Lorenz-Online. There, an interactive instance of the notebook can be run in the cloud.





---

**Algorithm 1** Online learning algorithm for the L96 model. Bold-face $\mathbf{X}(\mathbf{Y})$ indicate vectors with all $K(J)$ elements

---

**Require:** Pretrained LR-parameterization $\mathcal{P}_\theta$ with parameters $\theta$

**Require:** Initial conditions $\mathbf{X}_0$ and $\mathbf{Y}_0$

**Require:** Two-level HR model with time step $\Delta t_{\mathrm{HR}}$

**Require:** One-level LR model with time step $\Delta t_{\mathrm{LR}} = N\Delta t_{\mathrm{HR}}$

**Require:** Nudging time scale $\tau_{\mathrm{nudging}}$

**Require:** Feature memory $\mathcal{F}$ and target memory $\mathcal{T}$

**Require:** Training frequency $M$; learning rate $\alpha$ and batch size $m$

    Initialize HR with $\mathbf{X}_0$ and $\mathbf{Y}_0$; initialize "LR" with $\mathbf{X}_0$

    **for** $t = 1,...$ **do**

        Store difference at beginning of time step $\Delta\mathbf{X} = \mathbf{X}_{\mathrm{LR}} - \mathbf{X}_{\mathrm{HR}}$

        Store LR state at beginning of time step $\hat{\mathbf{X}}_{\mathrm{LR}} = \mathbf{X}_{\mathrm{LR}}$

        Store HR state at beginning of time step $\hat{\mathbf{X}}_{\mathrm{HR}} = \mathbf{X}_{\mathrm{HR}}$

        **for** $n = 1, N$ **do**

            Integrate HR model with forcing term $\Delta\mathbf{X}/\tau_{\mathrm{nudging}}$ added to the RHS of Eq. 2 to get new $\mathbf{X}_{\mathrm{HR}}$

        **end for**

        Compute "assumed" HR increment $\Delta\mathbf{X}_{\mathrm{HR\text{-}internal}} = \mathbf{X}_{\mathrm{HR}} - \hat{\mathbf{X}}_{\mathrm{HR}} - \Delta\mathbf{X}/\tau_{\mathrm{nudging}} * \Delta t_{\mathrm{LR}}$

        Store $\hat{X}_{\mathrm{LR},k}$ for $k \in 1,\ldots,K$ in $\mathcal{F}$

        Integrate LR model without parameterization term to get new $\mathbf{X}_{\mathrm{LR}}$

        Store $\Delta\mathbf{X}_{\mathrm{HR\text{-}internal}} - (\mathbf{X}_{\mathrm{LR}} - \hat{\mathbf{X}}_{\mathrm{LR}})$ for $k \in 1,\ldots,K$ in $\mathcal{T}$

        Update LR state $\mathbf{X}_{\mathrm{LR}} \leftarrow \mathbf{X}_{\mathrm{LR}} + \mathcal{P}_\theta(\hat{\mathbf{X}}_{\mathrm{LR}})$

        **if** $t$ mod $M = 0$ **then**

            Optimize loss for samples in $\mathcal{F}$ and $\mathcal{T}$: $\mathcal{L}_\theta = (\mathcal{P}_\theta(\mathcal{F}) - \mathcal{T})^2$ using stochastic gradient descent with learning rate $\alpha$ and batch size $m$

            Empty $\mathcal{F}$ and $\mathcal{T}$

        **end if**

    **end for**

---

Algorithm 1 outlines the workflow for coupled learning in the L96 framework. There are several hyper-parameters. First, the

time steps $\Delta t_{\mathrm{HR}}$ and $\Delta t_{\mathrm{ML}}$. In the easiest case, they are the same. However, more realistically, the HR model has a finer time step than the ML-LR model. For the experiments here, I used $N = 10$, i.e. $\Delta t_{\mathrm{ML}} = 0.01$.

The experiments indicate that coupled learning works well in both cases. One slight difference is that the learned linear regression intercept parameter $b$ is slightly different from the reference in the case where the HR time step is smaller. This is likely an indication that the *linear superposition assumption* during the HR integration is not perfect. However, the differences

are very small.

Another hyper-parameter is the update frequency of the neural network $M$. The experiments show that updating every time step causes the parameters to change a lot every update step. This is likely because the batch, which has size $K$, is only a small sample of the parameter space that is also potentially correlated. To combat this, we can gather the features and targets

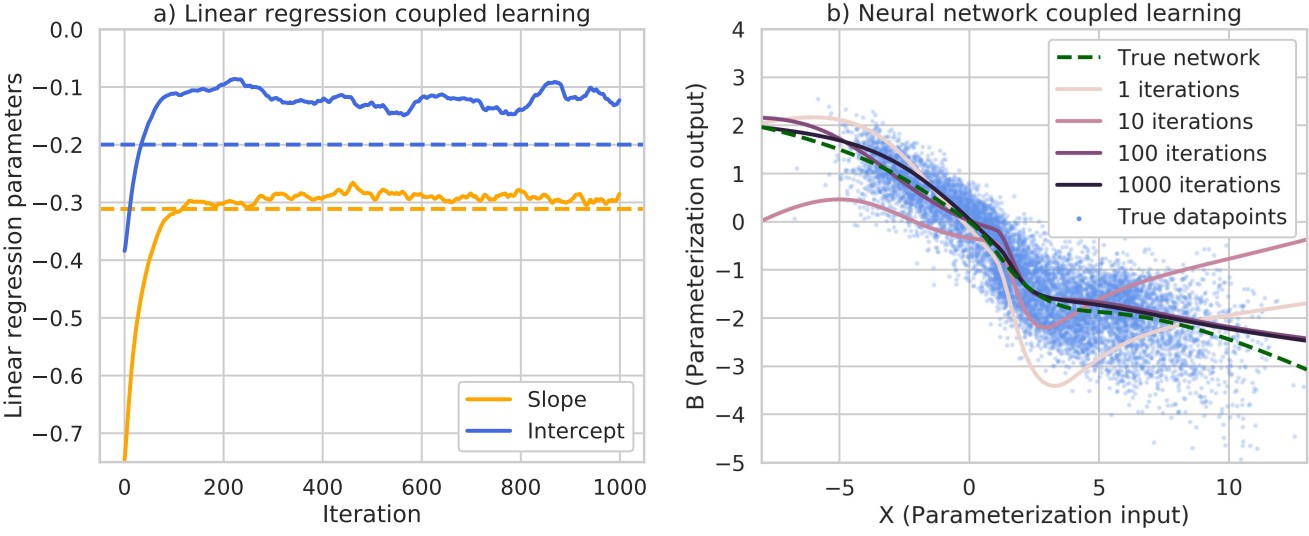

**Figure 4.** (a) Evolution of linear regression parameters $a$ (slope) and $b$ (intercept). An iteration on the x-axis corresponds to one gradient descent update which in this case is equal to ten ML-LR model time steps. (b) Evolution of the neural network parameterization starting with the "wrongly" pretrained fit. See the Jupyter notebook for an animated version of this.

over several ML-LR model time steps before doing an update step. Here, I used $M = 10$. This results in significantly smoother parameter convergence. Another potential advantage of updating only every few time steps is that the ML model can evolve more freely, thereby covering a larger fraction of the state space.

Finally, the nudging time scale $\tau_{\text{nudging}}$ has an impact on the fit. When the time scale is equal to the time step of the LR model, the HR model will be fully pulled towards the LR state. This works well for the L96 model but for complex ESMs this nudging will likely throw the HR model too much off its attractor. Weaker nudging, however, means that the HR and LR states at the beginning of the time step are further apart, which introduces an error. If the nudging time scale is too large, the ML targets will eventually become meaningless (see sensitivity experiments in the accompanying Jupyter notebook). For the experiments shown in Fig. 4, $\tau_{\text{nudging}} = 0.1 = 10\Delta t_{\text{LR}}$.

The same algorithm can be used to learn much more complicated parameterization such as a neural network (Fig. 4b). The $X-B$ curve gradually approaches the one learned *offline*. One final note on the L96 experiments: I did not exhaustively search for the best combination of hyper-parameters because the L96 experiments mainly serve as proof-of-concept. For coupled learning in a real modeling setup, the parameters are likely very different.

## 5 Algorithms for online learning in the super-parameterization and 3D HR frameworks

The fact that the method works in the L96 setup is a comforting sanity check. However, L96 does not exhibit any of the issues that require an coupled learning approach in the first place: an *offline* parameterization for the L96 model is stable and



does not show major biases. In this section, I will outline how coupled learning algorithms can be applied to 3D CRMs and super-parameterized GCMs.

### 5.1  3D high-resolution models

The 3D HR case is similar to the L96 setup. The key difference is that the scale separation is not clearly defined as in L96 or SP but rather downscaling (coarse-graining) and upscaling is required to get the HR state on the LR model grid and, reversely,
apply the forcing term, which is computed on the LR model grid, in the HR model. Issues with this will be further discussed in Section 6. The other difference between algorithms 1 and 2 is the way the gradient update is computed. In the L96 case the features and targets are stored in memory. This is unpractical for the HR setup since it requires storing several 3D fields over several time steps. Rather, in algorithm 2 the gradients are computed directly at each time step and collected in a single gradient vector $\mathcal{G}$, which is then used to update the parameters every $M$ steps. This also allows computing the gradients locally on each
node and then collecting them. The size of $\mathcal{G}$ is equal to the number of network parameters and, therefore, manageable. There is also no explicit batch size in this version of the algorithms. Rather, the batch size is implicitly given by $M \times K$.

One major conceptual difference of the 3D HR case to SP (see below) lies in what is actually learned by the neural network during coupled learning. In SP, the CRM is purely responsible for clouds and turbulence while a 3D HR model also evolves globally according to its own set of physics. What this means is that the neural network essentially learns a sub-grid correction
term that compensates for everything(!) missing from the LR model dynamics and non-ML physics in comparison to the HR model (HR$'$ → LR$''$ in Fig. 2). So even if all parameterizations except for convection are present in the LR, the network will not only learn convective tendencies. On the one hand, this is exactly what is required to get the LR closer to the expensive HR simulation. On the other hand, this makes the interpretation of what the network does a little more complicated.

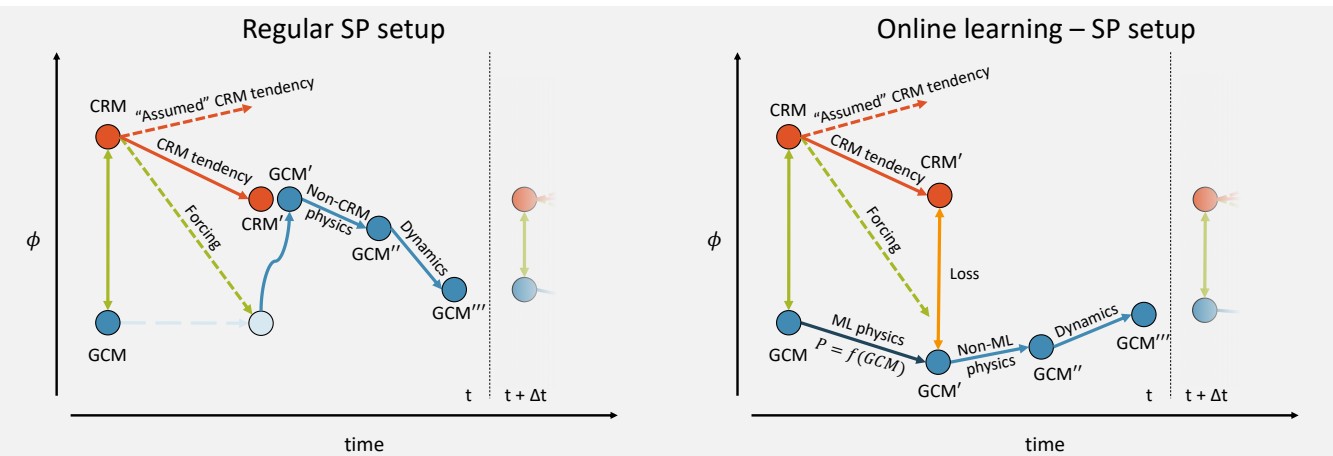

**Figure 5.** Evolution of a tracer $\phi$ during a regular SP step on the left and for coupled learning on the right.





---

**Algorithm 2** Online learning algorithm for 3D HR models. Bold-face vectors indicate state vectors for all grid columns $k \in 1, \ldots, K$. Overbars denote vectors on the coarse LR model grid.

---

**Require:** Pretrained ML-parameterization $\mathcal{P}_\theta$ with parameters $\theta$

**Require:** Initial conditions on the HR grid $\mathbf{x}_0$

**Require:** Downscaling and upscaling algorithms $\mathcal{D}$ and $\mathcal{U}$

**Require:** HR model with time step $\Delta t_{\text{HR}}$

**Require:** LR model with time step $\Delta t_{\text{LR}} = N \Delta t_{\text{HR}}$

**Require:** Nudging time scale $\tau_{\text{nudging}}$

**Require:** Gradient memory $\mathcal{G} = 0$

**Require:** Training frequency $M$; learning rate $\alpha$

    Initialize HR with $\mathbf{x}_0$; initialize LR with $\bar{\mathbf{x}}_0 = \mathcal{D}(\mathbf{x}_0)$

    **for** $t = 1,\ldots$ **do**

        Store delta at beginning of time step $\Delta \bar{\mathbf{x}} = \mathcal{D}(\mathbf{x}_{\text{HR}}) - \bar{\mathbf{x}}_{\text{LR}}$

        Store LR state at beginning of time step $\hat{\mathbf{x}}_{\text{LR}} = \bar{\mathbf{x}}_{\text{LR}}$

        Store HR state at beginning of time step $\hat{\mathbf{x}}_{\text{HR}} = \mathbf{x}_{\text{HR}}$

        **for** $n = 1, N$ **do**

            Integrate HR with nudging term $-\mathcal{U}(\Delta \bar{\mathbf{x}})/\tau_{\text{nudging}}$ to get new $\mathbf{x}_{\text{HR}}$

        **end for**

        Compute "assumed" HR increment $\Delta \bar{\mathbf{x}}_{\text{HR-internal}} = \mathcal{D}(\mathbf{x}_{\text{HR}} - \hat{\mathbf{x}}_{\text{HR}} - \mathcal{U}(\Delta \bar{\mathbf{x}})/\tau_{\text{nudging}} * \Delta t_{\text{LR}})$

        Integrate LR model (only dynamics and non-ML physics) to get intermediate $\bar{\mathbf{x}}_{\text{LR}}$

        Compute target $\mathcal{T} = \Delta \bar{\mathbf{x}}_{\text{HR-internal}} - (\bar{\mathbf{x}}_{\text{LR}} - \hat{\mathbf{x}}_{\text{LR}})$

        Compute loss for each column $\mathcal{L}_{\theta,k} = (\mathcal{P}_\theta(\hat{x}_{\text{LR},k}) - \mathcal{T}_k)^2$

        Store gradient: $\mathcal{G} \leftarrow \mathcal{G} + \frac{1}{M} \frac{1}{K} \sum_k \nabla_\theta \mathcal{L}_\theta$

        Add ML increment: $\bar{\mathbf{x}}_{\text{LR}} \leftarrow \bar{\mathbf{x}}_{\text{LR}} + \mathcal{P}_\theta(\hat{\mathbf{x}}_{\text{LR}})$

        **if** $t \bmod M = 0$ **then**

            Update parameters $\theta$ using gradients $\mathcal{G}$ with rate $\alpha$

            $\mathcal{G} = 0$

        **end if**

    **end for**

---

## 5.2 Super-parameterization

Similar to L96, SP has the advantage of a clean scale-separation, which makes the parameterization learning task easier. It also provides a good framework for coupled learning since SP already has the LR model and the embedded CRMs running in parallel. Because the embedded CRMs do not have any large-scale dynamics on their own, the time step schematic in Fig. 5 looks different to Fig. 2. In contrast to regular SP, the LR model state is not set to the CRM state after the CRM integration. Instead, the LR model evolves on its own according the the ML physics and the difference between CRM′ and LR′ is the loss to



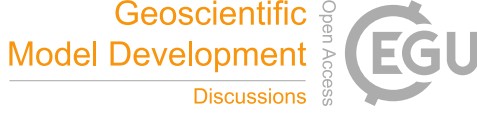

---

**Algorithm 3** Online learning algorithm for super-parameterized LR models. This algorithm is specific to the SP-CAM code structure. Note that the notation is slightly different from algorithm 2: the LR model state $\mathbf{x}$ now does not have an overbar and $\bar{x}_{\mathrm{CRM}}$ denotes the averaged CRM state.

---

**Require:** Pretrained ML-parameterization $\mathcal{P}_\theta$ with parameters $\theta$

**Require:** Initial conditions $\mathbf{x}_0$

**Require:** Embedded SP-CRM with time step $\Delta t_{\mathrm{CRM}}$

**Require:** GCM with time step $\Delta t_{\mathrm{GCM}} = N\Delta t_{\mathrm{CRM}}$, $N \in \mathbb{Z}^+$

**Require:** Gradient memory $\mathcal{G} = 0$

**Require:** Training frequency $M$; learning rate $\alpha$

  Initialize GCM with $\mathbf{x}_0$

  Uniformly initialize each CRM grid column from $\mathbf{x}_0$

  **for** $t = 1,...$ **do**

    Call CRM but do not update GCM state; internally this computes and applies the forcing term.

    Loss $\mathcal{L}_{\theta,k} = (x_{\mathrm{GCM},k} + \mathcal{P}_\theta(x_{\mathrm{GCM},k}) - (\bar{x}_{\mathrm{CRM}})_k)^2$

    Store gradient: $\mathcal{G} \leftarrow \mathcal{G} + \frac{1}{M}\frac{1}{K}\sum_k \nabla_\theta \mathcal{L}_\theta$ (can be done within each MPI thread)

    Add ML tendency: $\mathbf{x}_{\mathrm{GCM}} \leftarrow \mathbf{x}_{\mathrm{GCM}} + \mathcal{P}_\theta(\mathbf{x}_{\mathrm{GCM}})\Delta t_{\mathrm{GCM}}$

    **if** $t \bmod M = 0$ **then**

      Globally collect gradients $\mathcal{G}$

      Update parameters $\theta$ using gradients $\mathcal{G}$ with rate $\alpha$

      $\mathcal{G} = 0$

    **end if**

  **end for**

---

minimize. Algorithm 3 describes coupled learning specifically for SP-CAM. The interactions between the LR model and CRM are already contained in the CRM function call. This means that only few changes to the code are required: the neural network forward and backward passes have to be implemented, in addition to the optimizer and the communication of the gradients between the threads.

## 6  Discussion

### 6.1  Which variables have to be forced/predicted by the neural network?

In the three original ML parameterization studies, of the prognostic variables, only temperature and humidity were used in the input and output. This was done to reduce the complexity of the problem to the fewest prognostic variables necessary to produce a general circulation. In coupled learning, the variables used by the ML parameterization also have to be forced in the HR model. The HR model will typically have many more prognostic variables but it is alright for those to evolve without

forcing. In fact, this might be necessary since the HR and LR models might have different prognostic variables. This is the





case in SP where only the LR model prognostic variables are forced during CRM integration. If the variables predicted by the neural network differ, for example temperature vs. moist static energy, an additional conversion step has to be added to the up- and downscaling described below.

So theoretically coupled learning should work fine even if only temperature and humidity are forced/predicted. However, there are reasons for going beyond this. First, it is likely that the network skill suffers from not having information about e.g. cloud water. We saw this in RPG18 where the network was essentially unable to produce a shallow cloud heating signature in the sub-tropics. Second, to implement physical constrains it is necessary to add more variables in order to close the conservation budgets, which we will discuss now.

### 6.2 Physical constraints

A major critique of machine learning and especially neural network parameterizations is that they do not obey physical constraints. However, Beucler et al. (2019) recently showed that it is possible to encode physical constraints in neural networks if the conservation equations are known. There are two ways of doing so: First, violation of constraints can be added to the loss term during neural network training. This does not guarantee that the constraints are exactly obeyed, particularly outside of the training regime, but in practice might come close. The second method is to hard-code the conservation constraints into the last layers of the neural network. This ensures exact conservation and has been shown to only hurt the *offline* performance of the network slightly.

One downside of implementing physical constraints in Beucler et al. (2019) is that it requires predicting all prognostic variables that occur in the conservation budget equations. In effect, this increased the size of the output vector from 65 in RPG18 to 218. This now also includes variables that we might not actually care about like the snow storage term. Anecdotally, more variables also means more potential for things to go wrong, e.g. instabilities to develop. One possibility to reign in this complexity in *offline* and coupled learning is to omit some of these terms from the output vector and simply set them to zero in the budget equations. While this makes it impossible for the network to exactly reproduce the target (where all terms of the budget equation are used), this essentially forces the network to make the closest prediction to the target that lies on its own manifold of physically conserving solutions. If the omitted terms are small, this should still yield good results.

When using coarse-grained HR output as training data as in BB18, the residuals (Eq. 1) do not obey any conservation relations. In coupled learning, physical constraints could still be encoded however. All one needs to know is the budget equations valid on the LR model grid, i.e. the equations a traditional parameterization would also obey. The network will then learn the best physically conserving sub-grid correction term to bring the LR model closer to the HR model.

### 6.3 Up- and downscaling

Another issue is how to convert 3D fields from the LR model to the HR grid and vice versa. I already mentioned downscaling or coarse-graining along with some issues in the context of discussing BB18. For coupled learning in the 3D HR setup (Algorithm 2) a downscaling algorithm $\mathcal{D}$ is required to transform the HR state $x_{HR}$ to the LR model grid to compute the ML targets. Upscaling $\mathcal{U}$ is used to apply the forcing term, which is computed on the LR model grid, in the HR model. The





simplest method for downscaling is to simply average the HR values onto the LR model grid and interpolate if necessary. In signal processing this is the equivalent of applying a rectangular filter which potentially leads to aliasing. It might be worth investigating common filtering methods, such as using a Gaussian low-pass filter. [10] For upscaling, simply taking the LR model grid value that corresponds geographically to each HR grid point will results in sharp boundaries for the HR forcing field. A different way would be to use a smoother interpolation function, for example a spline. In practice, how problematic sharp boundaries in the forcing would be is hard to say without trying it out.

## 6.4 Technical challenges

Depending on the setup, there are some daunting technical challenges for the implementation of coupled learning. SP-CAM represents the easiest case because it already has the embedded CRMs running in parallel with the LR model with coupling. The key challenge here would be the implementation of the neural network forward and backward pass. We have already implemented the forward pass in RPG18 by hard-coding it in Fortran. This works but is error-prone, hard to debug and cumbersome. Backpropagation along with a modern gradient descent algorithm like Adam (Kingma and Ba, 2014) would add to the complexity. Another option is to call Python from Fortran[11] but this is potentially slow. Further, since the network parameters are global, the gradient descent step has to happen globally as well requiring communication between the nodes. The Python-Fortran interface currently is a major obstacle in ML parameterization research that begs for a simpler solution.[12].

For the 3D HR setup, in addition to the neural network implementation and the up-/downscaling issues, coupled learning requires two models to be run in parallel communicating every few time steps. This potentially requires quite a lot of engineering. My guess is that a successful and relatively quick implementation of coupled learning requires extensive working knowledge with the atmospheric models used.

## 6.5 How efficient is the online learning algorithm?

Running a HR model is expensive. Therefore, it is essential that the coupled learning algorithm is efficient enough to learn from a limited number of coupled HR simulations. To judge this, L96 is a bad toy model because it is so far removed from the actual problem. On the one hand, the parameterization task is exceedingly easy (one input, one output). On the other hand, it has 32 "LR" grid points while a 2-degree global LR model has more than 8,000, yielding a much larger sample for each gradient descent update. Further, there are a large range of hyper-parameters to tune. For a dry run, one could use a network trained *offline* on a reference dataset and then simulate coupled training by using a different, non-shuffled dataset (e.g. the +4K run from RPG18). This should provide guidance for choosing hyper-parameters and give a rough estimate of how many iterations are required.

---

[10]See https://dsp.stackexchange.com/questions/6313/low-pass-filter-parameters-for-image-downsampling for a related discussion.

[11]see Noah Brenowitz's blog post: https://www.noahbrenowitz.com/post/calling-fortran-from-python/

[12]CLIMA might be just that eventually: https://github.com/climate-machine/CLIMA; or alternatively the Sympl and CliML frameworks (Merwin Monteiro et al., 2018)



## 7   Conclusions

Coupled learning is a potential method to combat some of the main obstacles in ML parameterization research: instabilities and
tuning. In this paper my aim was to present the algorithms and challenges as clearly as possible and demonstrate the general
feasibility in the L96 case. The next step will be to test coupled learning in a more realistic framework. Some open questions
are: How much weight should be given to new samples, particularly if the tendencies are substantially chaotic? Are the HR
and ML-LR model guaranteed to converge? Will the *linear superposition assumption* break down if the forcing becomes too
large? How should situations be handled where the model crashes after all? Finally, coupled learning can only fix short term
prediction errors, which raises the question to which degree this would lead to a decrease in long-term biases.

There are a number of problems with ML parameterizations that coupled learning cannot address. First and foremost for cli-
mate modeling: generalization, i.e. the ability of a neural network parameterization to perform well outside its training regime.
Neural networks are essentially non-linear regression algorithms and should not be expected to learn anything beyond what
they have encountered during training (Scher and Messori, 2019). The research area of learning physical laws with deep learn-
ing is still in its infancy. For this reason Schneider et al. (2017a) advocate sticking to physically motivated parameterizations
and improve the tuning process. Note that coupled learning can still be used to tune parameters in existing parameterizations if
they are coded up in differentiable fashion.

      Another issue unsolved by coupled learning is stochasticity. Any deterministic ML model that minimizes a mean error will be
unable to represent random fluctuations in the training dataset. This leads to smoothed out predictions. The case for stochastic
parameterizations has been growing steadily (Berner et al., 2015; Palmer, 2019) raising the question how stochasticity can
be incorporated into ML parameterizations. Two possible approaches could be using generative adversarial networks (GANs;
Subramanian et al., 2018) or using a parametric distribution.[13] How to combine coupled learning with GANs, however, is not
readily apparent.

      Finally, high-resolution HR models might be better than coarse LR models but they still are not the truth. Our best knowledge
of the true behavior of the atmosphere comes from observations. The problem is that observations are intermittent in space and
time and, in the case of remote sensing, indirect. So how to learn from such data? Schneider et al. (2017a) propose a parameter
estimation approach using an ensemble Kalman inversion,a gradient free method for parameter optimization (Garbuno-Inigo
et al., 2019). The second best guess of the truth are re-analyses, such as the ERA5[14] dataset, which provides 3D fields every
3 hours. It could well be worth spending some thoughts on exploring how re-analyses could be used for ML parameterization
training.

Clouds are incredibly complex. No wonder then that we humans have such trouble shoving them into mathematical concepts.
We need any assistance we can get. Could ML provide us with such? The verdict is still out. First studies show that ML models
are, in general, capable of representing sub-grid tendencies but the way towards actually improving weather and climate

---

[13]Parametric approaches have been commonly used for post-processing of NWP forecasts (Rasp and Lerch, 2018), however mostly for single output tasks.
Realistic multi-variate predictions need to take into account covariances, which might require further research.

[14]https://www.ecmwf.int/en/forecasts/datasets/reanalysis-datasets/era5





models poses several obstacles. Coupled online learning could be one potential solution out of many to overcome some of these obstacles.

*Code availability.* All code (version 1.0) is available here: https://github.com/raspstephan/Lorenz-Online. The L96 experiments are all contained in a single Jupyter notebook which anyone can launch and interact with here: https://mybinder.org/v2/gh/raspstephan/Lorenz-Online/master?filepath=coupled-learning.ipynb

*Competing interests.* The author declares no competing interests.

*Acknowledgements.* I thank Chris Bretherton, Noah Brenowitz, Tapio Schneider, Sebastian Scher, David John Gagne, Tom Beucler, Mike
Pritchard and Pierre Gentine for their valuable input. I acknowledge funding from the German Research Foundation, partly through the project SFB/TRR 165 "Waves to Weather."





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
