# Peer review of "Coupled online learning as a way to tackle instabilities and biases in neural network parameterizations: general algorithms and Lorenz 96 case study (v1.0)"

_Geoscientific Model Development, 2019_

## Referee Comment (RC1) · Anonymous Referee #1 · 20 Jan 2020

This manuscript describes a method which hybridises machine learning with traditional numerical methods for simulating the Earth System that could avoid issues of numerical stability that impacted earlier attempts. The idea is to run a high-resolution model in parallel with the low-resolution, machine learning-hybridised model and to repeatedly retrain the machine learning algorithm based on how the high-resolution model evolves. It is essential to keep the high-resolution model synchronised to the low-resolution model through nudging, so that both are simultaneously situated in equivalent regions of their respective attractors. The idea is demonstrated in a toy model context, with linear regression and a neural net, and is also discussed for a real Earth System model. In the latter case, the author suggests how this technique could be applied when the

"high-resolution" model is literally a high-resolution configuration of the low-resolution model, but they also discuss how existing superparametrized models could be taken advantage of.

Although the technique is shown to work well for a toy model, it is unknown whether it can also work in a realistic setup. On this I have no strong intuition either way, and I can't think of any obvious reasons why it wouldn't work. I therefore enthusiastically recommend the manuscript for publication, subject to the corrections below which are mostly editorial.

**1 Minor comments**

- 32: "So far, three studies..." Chevallier at ECMWF conducted a number of studies on radiation parametrization 20 years ago which as far as I'm aware also implemented all three steps. See "Use of a neural-network-based long-wave radiative-transfer scheme in the ECMWF atmospheric model", Chevallier et al. , QJRMS (2000)

- 240 (whole paragraph): This doesn't necessarily warrant a change to the manuscript, but reading this paragraph made me think of the incremental 4D-Var algorithm used for data assimilation at various NWP centres, including ECMWF. There too, one needs to frequently interpolate between high- and low-resolution grids. This is for "online" use, not simply postprocessing. The innovations (observation - background) are computed using a high-resolution nonlinear model. These are then interpolated to a low-resolution grid so that the cost function gradient can be computed with low-resolution tangent-linear and adjoint models during minimisation. The resulting analysis increment is then upscaled for application to the high-resolution model, and so on. The fact that they successfully use high-/low-resolution models with a difference of $\sim$5 in grid-spacing makes me not so

concerned about the issues the author discusses. Though admittedly, I have no idea how they actually do the interpolation.

**2 Technical corrections**

- 3: "a machine learning algorithm"

- 4: "the trained algorithm was"

- 31: "and is coupled to the"

- 64: "second fix involves a multi-time-step"

- 77: "since they do not"

- Figure 2: $\Delta\phi_{\text{nudging}} = \Delta t_{\text{LR}}\Delta\phi/\tau_{\text{nudging}}$

- Equation 2: define overbar (especially since it is redefined in Algorithm 2)

- Equation 3: $(Y_{j+2,k} - Y_{j-1,k})$

- Algorithm 1: "Store $\Delta\mathbf{X}_{\text{HR-internal}}$..." — I think these should be nonbold with a $k$ subscript

- 150: "$\Delta t_{\text{HR}}$ and $\Delta t_{\text{LR}}$" — I guess "$\Delta t_{\text{ML}}$" was a typo

- 157: "because the batch, which has size $m$", no?

- Figure 4: To make the reading more smooth, please refer to this figure in the text, for example on line 152

- 168: It took me a while to understand this sentence, until I replaced "learn" with "train". Then it made perfect sense. I recommend you make this change

- 168: "a much more complicated"

- 169: a clarification — "gradually approaches the one learned offline *using the training data generated with the correct parameters $F$, $h$, $c$ and $b$.*" I hope I've understood this correctly...

- 174: "issues that require a coupled"

- 181: this is, as far as I can see, the first mention of Algorithm 2. Please add a brief description, on line 178, for example

- Figure 5: Again, I couldn't find a reference to this in the text. Please add one

- 200: I had to look up the meaning of "SP-CAM". Please define the acronym here

- 217: "physical constraints"

- 225: I don't understand "offline performance". Do you mean the computational performance is slower because you need to add extra operations to ensure conservation? If so why does this affect offline but not online performance? Or do you mean the technique was only tested offline and was found to be less accurate (performance = accuracy then) than with no conservation checking?

- 293: I don't think the HR and LR acronyms are necessary here

- 297: "best guess of the truth would be a reanalysis"

---

## Referee Comment (RC2) · Anonymous Referee #2 · 27 Jan 2020

This paper address an interesting approach to perform ML-based subgrid parameterization. The method itself is relevant for a publication, but I still think that some points could be discussed further.

General comments:

- I feel some inadequate between the general objective of the method the setup chosen to address this objective. Even the author acknowledges clearly that L96 model is not adequate, it is difficult for me to have a clear idea of what was or not shown by the L96 example. Maybe it can be summarize in the form of an introduction (or conclusion) to section 6:

1) questions addressed by online learning and that L96 confirms (mainly sanity check as far as I understand, but it shows also that you could improve an existing ML configuration)

2) question addressed by online learning and for which L96 is not sufficient and would require more test (e.g. improvement of a forecast of the ML-LR model without HR)

3) question not addressed by online learning as introduced in the paper but that could be implemented (e.g. physical constraints)

4) question not addressed yet by online learning (e.g. real observations, stochastic parametrizations)

I think such a summarized section would help the reader to clarify the contributions of the paper.

- I have also some concern about the justification of the nudging. The nudging aims at avoiding HR and LR to diverge as expected in case of chaotic models. So, my understanding is that it aims at correcting from errors due to initial conditions errors (which are due to past errors themselves). But then, could you explain more in what way restoring the HR model to the LR state could be problematic? You said that it was due to the fact that you can be outside the attractor, but isn't the purpose of ML-LR to be precisely in the HR attractor? Or you mean that you should nudge carefully for the first training step before ML-LR has converged toward the HR attractor? If there is a persistent bias between HR and ML-LR, it means either that the Neural Net is not able to learn it or that you have corrected it by the nudging, so you don't tell the Neural Net to learn it. These are quick questions (maybe not all relevant), but I think a more extensive discussion about this point could be done.

- Last point: as far as I understand, there are two methodological novelties. First, the fact that you learn online (meaning that the LR forecast is already computed using the ML parametrization). Second, the way you construct your training set is a bit different

from all the methods you detailed in the introduction. Computing, for each time step, an "assumed" HR tendency could be done offline using the Non-ML physics and dynamics as a LR model. It would be different either from RPG18 in which there is no HR model per se, and it would be also different from BB18 in which they don't run a LR model but using a coarse-grained HR model. The second innovation is not presented as a specific point (maybe in a sense, it is equivalent SP setup), but it would worse a discussion. In other words: in Fig.2, if I remove the ML physics, and I make the training offline (afterwards), is it equivalent to a previous approach (and if not, would it be valuable)? In still other words: if the training frequency M is very large, what does it give?

Other details comments:

L47: "In our subsequent online tests", could you be more explicit? What are the tests you are referring to?

Eq. (1): even if is implicit could you quickly define all the terms of the Equation: $\overline{v}$, $\overline{\phi}$, $\overline{\nabla}$

L76: "Random forests are also competitive with neural networks for many types of ML problems". I find this assertion a bit too general to be really informative. Is there a particular problem, with a similar degree of complexity as subgrid parameterization, in which RF has similar performances as Neural Networks?

L107-109: HR increment calculation. Even if I support the effort to make the explanation not too technical, I think the description could benefit for a bit more of formalism. If you define what you call tendency and increment using equations, you could introduce very clearly the different terms. Also, you could add the time step to the formulas ($\Delta\phi_{\rm nudging}$) depend on a time step. Is the nudging term assumed constant between t and $t+\Delta_{LR}$?

L110: I don't really get here what is called "assumed HR-internal increment". What do you mean exactly by "the nudging and the HR-internal evolution are independent"?

L110: I think that using time reference would clarify a lot the expression. (I understand HR' is calculated at time t+\Delta t and HR is calculated at time t). Also; in theory, you can compute the increment of the HR model with or without nudging. So why do you need to make the "linear superposition assumption?"

L115: this is true if you neglect the error due to initial conditions, but what is the effect of such error on the loss?

L140: I am a bit surprised by the architecture. Several studies (Pathak et al. 2018, Dueben and Bauer 2018, Bocquet et al. 2019) suggest that it is relevant to take into account locality of the parametrization (see also Fig.3 of Wilks 2005 that suggests a quick spatial decorrelation). Could you comment on that?

L145: Here again, could you add a quick justification of the setup. In the introduction, the case you address is that the attractor of the pure LR-model and the ML-LR model is different. Is it not the case here without changing model configuration?

L158: Would it also be possible to add regularization term on the weights to avoid them to have a too strong update? Figure 4: Could you define the true network? If the true network is the one learnt offline with the real tendency, it seems to me that it is exactly the network you could find in an offline approach and for which you say it was subject to divergence. L209-210: I don't understand this point

On the code (found on GitHub): Learning linear model using Adam is not standard and probably not as efficient than using dedicated fit for linear problems.

very minor details:

L64: "an multi-time-step" -> a multi-time-step

P3 Footnote 2: could you expand SPCAM notation

L81: "ML-LR": Could you define the acronym the first time you are using it?

L121-122:" Here, I use the model as described in Schneider et al. (2017a)." the reference to Schneider is not the most relevant want to introduce the Lorenz model. This configuration is pretty standard and you find the same in numerous articles. Potentially, it is a good place to cite the already mentioned Lorenz 95 paper.

L135: It would be interesting to introduce the polynomial parametrization as in Wilks 2005, as it is rather standard for this model.

Algoritmh 3: SP-CAM -> SPCAM?

L200: SP-CAM (SPCAM) why is it specific to SPCAM?

---

## Author Comment (AC1) · 24 Mar 2020

I would like to thank the two referees for their constructive and thoughtful reviews. I tried to address their concerns to the best of my ability. I certainly believe that the manuscript is stronger after these modifications. My replies to the comments are listed below in *italics*.

[Figure]

Anonymous Referee 1

This manuscript describes a method which hybridises machine learning with traditional numerical methods for simulating the Earth System that could avoid issues of numerical stability that impacted earlier attempts. The idea is to run a high-resolution model in parallel with the low-resolution, machine learning-hybridised model and to repeatedly retrain the machine learning algorithm based on how the high-resolution model evolves. It is essential to keep the high-resolution model synchronised to the low-resolution model through nudging, so that both are simultaneously situated in equivalent regions of their respective attractors. The idea is demonstrated in a toy model context, with linear regression and a neural net, and is also discussed for a real Earth System model. In the latter case, the author suggests how this technique could be applied when the "high-resolution" model is literally a high-resolution configuration of the low-resolution model, but they also discuss how existing superparametrized models could be taken advantage of.

Although the technique is shown to work well for a toy model, it is unknown whether it can also work in a realistic setup. On this I have no strong intuition either way, and I can't think of any obvious reasons why it wouldn't work. I therefore enthusiastically recommend the manuscript for publication, subject to the corrections below which are mostly editorial.

*I also do not know whether this could work in a realistic setup. I hope to find out soon using CliMT as a more realistic model. Also thanks a lot for really looking at the algorithms in detail. I really appreciate the time you spent.*

Minor comments

32: "So far, three studies..." Chevallier at ECMWF conducted a number of studies on radiation parametrization 20 years ago which as far as I'm aware also implemented all three steps. See "Use of a neural-network-based long-wave radiative transfer scheme in the ECMWF atmospheric model", Chevallier et al. , QJRMS (2000)

*Thank you for pointing this out. I modified the text accordingly: "The first study to implement an online ML parameterization was done by Chevallier2000 who successfully emulated the ECMWF radiation scheme. More recently, three studies have implemented all three steps for moist convection in the context of an atmospheric model Brenowitz2018, OGorman2018, Rasp2018c."*

240 (whole paragraph): This doesn't necessarily warrant a change to the manuscript, but reading this paragraph made me think of the incremental 4D-Var algorithm used for data assimilation at various NWP centres, including ECMWF. There too, one needs to frequently interpolate between high- and low-resolution grids. This is for "online" use, not simply postprocessing. The innovations (observation - background) are computed using a high-resolution nonlinear model. These are then interpolated to a low-resolution grid so that the cost function gradient can be computed with low-resolution tangent-linear and adjoint models during minimisation. The resulting analysis increment is then upscaled for application to the high-resolution model, and so on. The fact that they successfully use high- /low-resolution models with a difference of âĹij5 in grid-spacing makes me not so concerned about the issues the author discusses. Though admittedly, I have no idea how they actually do the interpolation.

*Great point. I had not thought of this. I added this point to the paragraph: "Note also that up- and downscaling is done in operational data assimilation, for example 4DVAR,*

*where the adjoint model is run on a lower resolution."*

Technical corrections

*I fixed all of these.*

Anonymous Referee 2

This paper address an interesting approach to perform ML-based subgrid parameterization. The method itself is relevant for a publication, but I still think that some points could be discussed further.

*Thank you for urging me to clarify on some aspects of the paper. I hope that my revisions address your concerns.*

General comments:

- I feel some inadequate between the general objective of the method the setup chosen to address this objective. Even the author acknowledges clearly that L96 model is not adequate, it is difficult for me to have a clear idea of what was or not shown by the L96 example. Maybe it can be summarize in the form of an introduction (or conclusion) to section 6: 1) questions addressed by online learning and that L96 confirms (mainly sanity check as far as I understand, but it shows also that you could improve an existing ML configuration) 2) question addressed by online learning and for which L96 is not sufficient and would require more test (e.g. improvement of a

forecast of the ML-LR model without HR) 3) question not addressed by online learning as introduced in the paper but that could be implemented (e.g. physical constraints) 4) question not addressed yet by online learning (e.g. real observations, stochastic parametrizations) I think such a summarized section would help the reader to clarify the contributions of the paper.

*This is a good point. The purpose of using the L96 model is to demonstrate that the algorithm actually works. Without this demonstration I, and probably many readers, would have a much lower confidence that such a method could actually work in a "realistic" setup. In fact, I discuss your point 3) and the question of not having a HR model (2) in the discussions and your point 4) in the conclusions. However, I do agree that a summary of what the L96 model does and does not show would clarify things. I have now added such a discussion at the end of Section 4 in a subsection called "Purpose and limitations of L96 experiments". I believe that it makes most sense here but am happy to hear your thoughts.*

- I have also some concern about the justification of the nudging. The nudging aims at avoiding HR and LR to diverge as expected in case of chaotic models. So, my understanding is that it aims at correcting from errors due to initial conditions errors (which are due to past errors themselves). But then, could you explain more in what way restoring the HR model to the LR state could be problematic? You said that it was due to the fact that you can be outside the attractor, but isn't the purpose of ML-LR to be precisely in the HR attractor? Or you mean that you should nudge carefully for the first training step before ML-LR has converged toward the HR attractor? If there is a persistent bias between HR and ML-LR, it means either that the Neural Net is not able to learn it or that you have corrected it by the nudging, so you don't tell the Neural Net to learn it. These are quick questions (maybe not all relevant), but I think a more extensive discussion about this point could be done.

*Great question. I initially thought pulling the HR state fully towards the LR state would not be a problem, and it isn't in L96. However, first tests using a GCM showed that this leads to blow ups. The problem is that, yes, ideally ML+LR should be equal to HR but, especially in the beginning of training, they are not. Note also that it takes many time steps before the ML parameterization learns to compensate for the LR errors and never is fully able to do so (see Fig. 4). I would expect the neural net to NOT be able to learn a perfect correction ever, particularly in a system with chaos. I expanded my discussion of this point in the paper in section 4.3, also referencing common nudging time scales in literature.*

- Last point: as far as I understand, there are two methodological novelties. First, the fact that you learn online (meaning that the LR forecast is already computed using the ML parametrization). Second, the way you construct your training set is a bit different from all the methods you detailed in the introduction. Computing, for each time step, an "assumed" HR tendency could be done offline using the Non-ML physics and dynamics as a LR model. It would be different either from RPG18 in which there is no HR model per se, and it would be also different from BB18 in which they don't run a LR model but using a coarse-grained HR model. The second innovation is not presented as a specific point (maybe in a sense, it is equivalent SP setup), but it would worse a discussion. In other words: in Fig.2, if I remove the ML physics, and I make the training offline (afterwards), is it equivalent to a previous approach (and if not, would it be valuable)? In still other words: if the training frequency M is very large, what does it give?

*I thought about this comment for quite a while. I then ran an experiment to test your proposition. I turned off the ML parameterization and ran the algorithm to just collect the training data. Then I trained offline afterwards. What I found is that when the*

*nudging time scale is equal to the time step (which as I discuss in the paper might not work for CRMs) the collected training data is a good representation of the real data. With a weaker nudging, however, it is not anymore because the ML-LR and HR models drift too far away from each other. If the algorithm is run online, the two models will become closer and closer through iterative training. Without this, the algorithm might not work. I added a short discussion in the paper (section 4.3) and added the experiments in the notebook.*

Other details comments:

L47: "In our subsequent online tests", could you be more explicit? What are the tests you are referring to?

*I changed it to "When we subsequently implemented the ML parameterization in the climate model and ran it prognostically (online)". I hope this is clearer now.*

Eq. (1): even if is implicit could you quickly define all the terms of the Equation: $v, \phi, \nabla$

*Added.*

L76: "Random forests are also competitive with neural networks for many types of ML problems". I find this assertion a bit too general to be really informative. Is there a particular problem, with a similar degree of complexity as subgrid parameterization, in which RF has similar performances as Neural Networks?

*This statement was mostly based on experience. Since I couldn't find a good reference, I made the statement more specific: "Comparing the results of OGorman2018*

*to RPG18 or BB18, it also seems like random forests are competitive with neural networks for the parameterization problem."*

L107-109: HR increment calculation. Even if I support the effort to make the explanation not too technical, I think the description could benefit for a bit more of formalism. If you define what you call tendency and increment using equations, you could introduce very clearly the different terms. Also, you could add the time step to the formulas ($\Delta\phi_{\text{nudging}}$) depend on a time step. Is the nudging term assumed constant between t and t+

*Thanks for urging me to state things more clearly. I have done the following things. 1) I added a reference to Algorithm 2 where everything is defined in a lot of detail. 2) I defined increments and tendencies: "Note that "tendencies" are defined per unit of time, while "increments" are tendencies multiplied by a time step."*

Regarding the comment about the time step in the nudging. I think I do mention this in the following sentence: "The total increment from nudging then is $\Delta\phi_{nudging} = \Delta\phi/\tau_{nudging} * \Delta t_{LR}$."

*Yes, the nudging is constant. This is now also stated in the paper.*

L110: I don't really get here what is called "assumed HR-internal increment". What do you mean exactly by "the nudging and the HR-internal evolution are independent"?

*The HR model, which generally has a finer time, step evolves while being nudged towards the LR state. This means that strictly the evolution of the HR model is affected*

*by the nudging even if the nudging is subtracted, simply because the states change. In other terms: $dHR\_nudged - nudging! = dHR\_not\_nudged$. However, I am making this assumption so that I can get the not_nudged term that I want to learn by subtracting the nudging. I added this: "HR-internal evolution (i.e. the HR increment that would be in the absence of nudging)". I thought long and hard how to make this clearer but I have a hard time explaining it to people...*

L110: I think that using time reference would clarify a lot the expression. (I understand HR' is calculated at time t+$\Delta$ t and HR is calculated at time t). Also; in theory, you can compute the increment of the HR model with or without nudging. So why do you need to make the "linear superposition assumption?"

*I added a statement to refer to Fig 2 for notation. Hopefully this should clar up the time-step issues. 2) Yes, it would be possible to directly compute the "assumed" tendency, but to keep the states in sync you need the nuding. One would of course for each dt, run both versions of the CRM, nudged and non_nudged, but that would be a huge computational cost and effort. For this reason the algorithm is build the way it is.*

L115: this is true if you neglect the error due to initial conditions, but what is the effect of such error on the loss?

*Actually, the initial condition error is accounted for. If you look at Fig 2 you see that the two simulations already start out with different states at the beginning of the time step.*

L140: I am a bit surprised by the architecture. Several studies (Pathak et al. 2018, Dueben and Bauer 2018, Bocquet et al. 2019) suggest that it is relevant to take into account locality of the parametrization (see also Fig.3 of Wilks 2005 that suggests a

quick spatial decorrelation). Could you comment on that?

*I added this to the paper: "Studies (Wilks2005, Dueben2018, Pathak2018, Bocquet2019) suggest that non-local and stochastic parameterizations achieve better results. However, the focus here is on developing a learning algorithm rather than achieving building the best parameterization which is why I opted for the simplest setup."*

L145: Here again, could you add a quick justification of the setup. In the introduction, the case you address is that the attractor of the pure LR-model and the ML-LR model is different. Is it not the case here without changing model configuration?

*I am not sure I understand this comment. The HR and LR L96 models, i.e. with and without the Y equation, do have different attractors. Also adding a ML parameterization changes the attractor of the model. So this setup should mimic the real GCM problem, at least in this regard. Maybe you are referring to the pretraining of the ML parameterization using different L96 parameters. This is simply done to give the ML methods, particularly the neural network, a starting point, so that we don't have to train from scratch. What exactly the starting point is, should not matter that much.*

L158: Would it also be possible to add regularization term on the weights to avoid them to have a too strong update?

*Yes, this is possibly a good point. However, as mentioned I did not spend too much time tuning the hyperparameters of the L96 setup since it serves only as illustration.*

Figure 4: Could you define the true network? If the true network is the one learnt offline with the real tendency, it seems to me that it is exactly the network you could find in an offline approach and for which you say it was subject to divergence.

*This is one of the limitations of the L96 model. As stated in the paper and clarified by your comment above, the L96 model does not have the main issue that motivated me to develop this algorithm in the first place, namely unstable simulations when trained offline. So for L96 we can get the "true" parameterization. This would not be possible in the real GCM case.*

L209-210: I don't understand this point

*Typically CRMs have more prognostic variables than coarse GCMs, for example hydrometeors. These cannot be forced then but the algorithm allows for this. I made the statement in the paper clearer hopefully.*

On the code (found on GitHub): Learning linear model using Adam is not standard and probably not as efficient than using dedicated fit for linear problems.

*Yes, this is true but in order to learn online we need an incremental update algorithm. In fact, I am comparing the linear regression fits I get using Adam with the fit I get using a more traditional fitting technique (cell 30) and they match very well.*

very minor details:

*All of these were fixed unless otherwise noted.*

L135: It would be interesting to introduce the polynomial parametrization as in Wilks 2005, as it is rather standard for this model.

*While it would be interesting indeed, I don't think it would add to the paper. I chose the linear regression because it is easy to display the evolution of the two parameters and therefore illustrates what the algorithm is doing. The neural network is chosen as a more generic ML method that also reflects what people want to do in real GCMs. A polynomial would be in-between LR and NN in complexity.*

L200: SP-CAM (SPCAM) why is it specific to SPCAM?

*It is specific to super-parameterized models. I changed the text accordingly.*